# SARS-CoV-2: A Master of Immune Evasion

**DOI:** 10.3390/biomedicines10061339

**Published:** 2022-06-07

**Authors:** Alberto Rubio-Casillas, Elrashdy M. Redwan, Vladimir N. Uversky

**Affiliations:** 1Biology Laboratory, Autlán Regional Preparatory School, University of Guadalajara, Autlán 48900, Jalisco, Mexico; 2Biological Science Department, Faculty of Science, King Abdulaziz University, P.O. Box 80203, Jeddah 21589, Saudi Arabia; lradwan@kau.edu.sa; 3Therapeutic and Protective Proteins Laboratory, Protein Research Department, Genetic Engineering and Biotechnology Research Institute, City for Scientific Research and Technology Applications, New Borg EL-Arab, Alexandria 21934, Egypt; 4Department of Molecular Medicine and USF Health Byrd Alzheimer’s Research Institute, Morsani College of Medicine, University of South Florida, Tampa, FL 33612, USA

**Keywords:** SARS-CoV-2, COVID-19, cell entry, evasion mechanisms, cell-to-cell fusion, cell-in-cell syncytia, nanotube, glycan capping, extracellular vesicles, exosomes

## Abstract

Viruses and their hosts have coevolved for a long time. This coevolution places both the pathogen and the human immune system under selective pressure; on the one hand, the immune system has evolved to combat viruses and virally infected cells, while viruses have developed sophisticated mechanisms to escape recognition and destruction by the immune system. SARS-CoV-2, the pathogen that is causing the current COVID-19 pandemic, has shown a remarkable ability to escape antibody neutralization, putting vaccine efficacy at risk. One of the virus’s immune evasion strategies is mitochondrial sabotage: by causing reactive oxygen species (ROS) production, mitochondrial physiology is impaired, and the interferon antiviral response is suppressed. Seminal studies have identified an intra-cytoplasmatic pathway for viral infection, which occurs through the construction of tunneling nanotubes (TNTs), hence enhancing infection and avoiding immune surveillance. Another method of evading immune monitoring is the disruption of the antigen presentation. In this scenario, SARS-CoV-2 infection reduces MHC-I molecule expression: SARS-CoV-2’s open reading frames (ORF 6 and ORF 8) produce viral proteins that specifically downregulate MHC-I molecules. All of these strategies are also exploited by other viruses to elude immune detection and should be studied in depth to improve the effectiveness of future antiviral treatments. Compared to the Wuhan strain or the Delta variant, Omicron has developed mutations that have impaired its ability to generate syncytia, thus reducing its pathogenicity. Conversely, other mutations have allowed it to escape antibody neutralization and preventing cellular immune recognition, making it the most contagious and evasive variant to date.

## 1. Introduction

Although at the beginning of the pandemic, the fatality in COVID-19 patients was lower (2.15%) than those of its nearest cousins, SARS-CoV-1 (9.5%) and MERS-CoV (34.4%), SARS-CoV-2 has a greater capacity for infecting people and can therefore cause greater global morbidity and fatality [1]. Because of the growing concern about the surge of new viral mutants that could disrupt governmental health interventions, decrease the effectiveness of vaccines or natural immune protection as well as antiviral treatments, public health monitoring organizations have undertaken an important initiative to utilize viral genetic data to trace pandemic growth [2].

The World Health Organization (WHO) has grouped rising SARS-CoV-2 variants into separate categories depending on their infectivity potential, with variants of concern (VOCs) requiring quick resolution, and various VOCs (Alpha, Beta, and Gamma) being closely monitored [3]. The Delta and Omicron variants are the two currently present VOCs, with Omicron exceeding Delta in terms of antibody resistance. Furthermore, recent work has discovered that the Omicron spike protein outperforms the spike of the Delta variant in terms of antibody evasion by up to 44 times, and has suggested that “most therapeutic antibodies will be ineffective against the Omicron variant and that double immunization with BNT162b2 might not adequately protect against severe disease induced by this variant” [4]. As a necessary repercussion, a wide genetic investigation and surveillance of SARS-CoV-2 were initiated to deal with the accelerated aggregation of virus genetic changes and to gain a better understanding of the virus’s evolutionary adaptability in humans in an attempt to produce better COVID-19 vaccines and therapeutic alternatives. [2].

Due to its proofreading exoribonuclease, the genetic code of SARS-CoV-2 was reported to acquire changes in two nucleotides over the course of a month, which is somewhat slower compared with other RNA viruses [5]. While the majority of accidental mutations are either silent, causing no modifications at the biological level, or harmful, compromising viral efficiency, others may provide a selection benefit; this results in their replication in succeeding viral populations, which have favorable characteristics and are frequently purified [2]. In circulating SARS-CoV-2 strains, aleatoric genomic changes were discovered, notably in the spike and nucleocapsid genomes, which are the most changeable genes in the viral genome [6]. Furthermore, confirmation of autonomous convergent alterations in the SARS-CoV-2 genetic code reveals that the virus is under constant and growing selection pressure at both the population and patient levels [2]. As global vaccination programs continue, an increasing percentage of inhabitants now have proper vaccine-induced immunity to the dominant virus, and this expanding level of protection is undoubtedly putting the virus under strong evolutionary pressure, leading to the emergence of variants capable of antibody escape [2].

According to recent investigations, the immune escape mutants have appeared and reappeared in chronic COVID-19 patients and immunocompromised individuals who are unable to successfully battle infection, resulting in the major alterations in the SARS-CoV-2 spike, as well as the proteins ORF1ab, ORF8, and nsp1 [7,8]. In addition to immunization, immunotherapies, including the antiviral Remdesivir and steroids, as well as convalescent plasma treatments and neutralizing immunoglobulins from recovered patients and antiviral monoclonal immunoglobulins, have been used to treat the COVID-19 disease [9,10].

Recent investigations have demonstrated that COVID-19 patients treated with convalescent plasma showed significant improvement in clinical symptoms, a reduction in the level of viral antigens, and an increase in the blood oxygen saturation and lymphocyte ratio [9], although the use of convalescent plasma also promotes the production of antibody escape variants [11,12]. As a result, these therapies can induce beneficial mutations in this virus. Because of fading or incomplete primary immunity, the use of inadequate antibodies in treatment with plasma from recovered patients and re-infection may create a selection pressure for immunological escape mutations [2]. The main evolutionary adaptation strategies of SARS-CoV-2 to avoid the immune system’s attack are described in this paper.

## 2. Strategies of Immune Evasion by SARS-CoV-2

There are at least seven reported strategies, which are utilized by SARS-CoV-2 for immune evasion:
Spike camouflage employs glycan molecules (epitope masking).Differential impairment of MHC-I-mediated antigen presentation by SARS-CoV-2 variants.SARS-CoV-2-driven inhibition of the interferon synthesis.SARS-CoV-2 induces incomplete mitophagy to avoid apoptosis of some infected cells and to increase virus replication.Cell–cell infection and immune evasion through cytoplasmic nanotubes.Cell–cell infection and lymphocyte cell death through syncytia formation.Immune evasion through exosome release.

### 2.1. Spike Camouflage Employing Glycan Molecules (Epitope Masking)

The genetic code of the SARS-CoV-2 virus was shared with the world by the Chinese CDC in January 2020 [13]. This triggered significant global efforts to develop an efficient vaccine, and while diverse platforms were used, they all utilized the spike protein’s genetic sequence, with the objective for the immunological system being to perceive the SARS-CoV-2 spike as a foreign antigen, stimulating the synthesis of specific antibodies against it, and therefore defending the organism from infection [14]. It is widely accepted that antibodies will attach to the spike’s specific receptor-binding domain (RBD), preventing the pathogen from infecting human cells [15].

Several viruses have adapted an epitope masking strategy by coating their envelope glycoproteins with glycans produced by the host, thereby preventing (or minimizing) antibody recognition. Numerous viruses have demonstrated the relevance of the N-associated glycosylation of envelope proteins for immunological escape, including hepatitis C virus [16,17], HIV-1 [18,19,20], hepatitis B virus [21], herpes simplex virus [22], and coronaviruses [23,24,25,26]. In addition to playing a crucial role in epitope camouflage, which represents an efficient method of immune evasion [24,27,28], the glycan barrier also enhances viral bonding, entrance, and membrane fusion [29].

Besides N-associated glycans, low levels of O-glycans in SARS-CoV-2 spike protein have recently been discovered [24,27,28,29]. While the oligomannose glycan content of SARS-CoV-2 (28%) is greater compared to normal host glycans [24], it is much lower when compared with the HIV-1 Env, which has 60% oligomannose-type glycans [20,30]. This shows that, in comparison with other viral glycoproteins (Figure 1), the SARS-CoV-2 spike protein contains fewer glycans, and they create a weaker protective layer that may be helpful for the attachment of the neutralizing antibodies [24].

It was pointed out that “SARS-CoV-2 infection is controlled by the opening of the spike protein receptor-binding domain (RBD), which transitions from a glycan-shielded ‘down’ to an exposed ‘up’ state to bind the human ACE2 receptor and infect cells” [23]. According to these researchers, the RBD is almost fully covered by the glycans in the ‘down state,’ offering a perfect camouflage. To allow ACE2 to attach to the receptor-binding motif (RBM), the RBD must shift from a ‘down’ to an ‘up’ state, and therefore the associated activation mechanism is required for cell entrance, but in addition, is also necessary for antibody recognition and neutralization [23].

In general, the viral RBD must switch from a ‘down’ to an ‘up’ state to attach to ACE2. However, the RBD of Omicron has trouble shifting to the ‘up’ state due to structural modifications caused by one of its numerous mutations, according to evidence recently published [31]. As a result, Omicron necessitates more ACE2 receptors than other variants to merge with host tissue. Considering cells from the lungs have considerably lower ACE2 numbers than cells in the nasopharynx, this could explain why Omicron does not readily infect lung tissue [31].

According to a very recent investigation, it was found that the Omicron variant features four novel spike mutations (S371L, N440K, G446S, and Q493R) that provide increased antibody resilience, with the most notable discovery being that the S371L mutation impaired monoclonal antibody neutralization in all four RBD classes [32]. Although the exact mechanism of antibody resistance is uncertain, computational modeling has indicated two options. First, changing Serine to Leucine in the RBD-down state obstructs the N343 glycan, perhaps modifying its structure and inhibiting class 3 antibodies that attach to this area. Second, in the RBD-up state, S371L may change the structure of the S371-S373-S375 loop, which may disrupt the attachment of class 4 antibodies that bind to a segment of this loop [32]. In a subsequent study, these researchers found a human monoclonal antibody that was able to block all sarbecoviruses tested and showed high affinity and potency in vitro and in vivo. The authors concluded that such an antibody is a suitable aspirant for outbreak contingency planning [33].

Recent studies show that glycosylation is essential for virus infectiousness. The spike protein of SARS-CoV-2 contains a unique furin cleavage region that promotes viral transmissibility and syncytia production in the infected cells. Representatives of the GALNT (N-acetyl-D-galactosamine (GalNAc) transferase) enzyme group regulate the addition of O-glycans around the furin cleavage region, resulting in reduced efficiency of furin cleavage and syncytia formation. Furthermore, it was discovered that the proline residue at position 681 is required for O-glycosylation. In vitro, mutation of this proline caused the GALNT1 glycosyltransferase function to be lost within the furin cleavage area [34]. P681 mutations in the highly contagious Alpha and Delta versions of SARS-CoV-2 prevent O-glycan addition, enhance furin cleavage, and boost the production of syncytia. Furthermore, GALNT group components capable of glycosylating the spike protein have been discovered in the human respiratory cells susceptible to SARS-CoV-2 infection. The presence/absence of O-glycans may impact virus transmission/tropism by modifying the furin cleavage of the spike. Therefore, these findings offer molecular insights that explain the relevance of the P681 mutations prevalent in Alpha and Delta variants [34].

Before the Delta variant, Alpha was the most contagious lineage of SARS-CoV-2. This was conceivable since, in contrast to the Wuhan strain, the Alpha variant had a histidine instead of proline at location 681 in the furin cleavage region in the spike protein at the S1/S2 juncture (residues ^681^PRRAR^685^). Furin cleavage needs an alkaline milieu, so in the Alpha variant, replacing proline (apolar) with histidine (positively charged) increased the furin cleavage efficiency, resulting in higher activity of spike proteins capable of infecting the cells. Furthermore, by substituting the proline residue, the Alpha variant eliminated the glycosylation site detected in the Wuhan strain [35]. This is significant, since carbohydrates obstruct the catalytic domain, preventing the protease from accessing it. The Delta variant also features a change at position 681, but instead of histidine, an arginine replaces the proline residue. Because it is an important amino acid, its mutation in the Delta variant facilitates spike protein cleavage, boosting cell identification and invasion, and resulting in higher infectiveness of this variant than that of the Alpha variant, thereby serving as one of the major factors for the enhanced infectiousness of the Delta variant [35].

The Omicron variant maintains the identical P681H substitution as the Alpha variant and contains a novel and exclusive glycosite, Threonine 376, discovered only in Omicron. In a 3D structure, this site is situated near a proline 681 residue, which regulates O-glycosylation [36]. A notorious increase in the usage of Core 2 type O-glycoforms was observed in the Omicron variant in comparison with the wild-type virus or its Delta variant, which is consistent with the addition of O-glycans [36]. This finding is transcendental because it was previously reported that, in vitro, mutation of this proline 681 caused GALNT1 glycosyltransferase function to be lost, as they regulate the addition of O-glycans around the furin cleavage region [34]. This suggests that the new Thr376 mutation, exclusive for Omicron, restored the capacity to add O-glycans that shield the furin cleavage site. As a consequence, a greater immune escape has been achieved by this variant, but with lesser pathogenicity, as will be discussed later.

### 2.2. Differential Impairment of MHC-I-Mediated Antigen Presentation by SARS-CoV-2 Variants

The clinical spectrum of COVID-19 appears to be distinct from that of SARS. Based on the epidemiological and clinical evidence, in comparison with SARS-CoV-1, SARS-CoV-2 infection has a prolonged incubation period (ranging from 0 to 24 days, with an average of 6.4), and pre-symptomatic individuals can transmit the virus to others [37,38]. Asymptomatic infections have been commonly recorded, putting community preventative mechanisms in jeopardy [37]. A significant proportion of recovered patients continue releasing viral elements in the upper respiratory system and digestive tract, necessitating a substantially longer hospitalization time [39,40,41,42]. Some COVID-19 patients showed significant viral RNA levels following hospital discharge [42]. Given the lack of synchronization between viral levels and clinical symptom progression, SARS-CoV-2 might have replicated widely in infected cells without being noticed by the antiviral response [43]. CTLs (cytotoxic T lymphocytes) are important players in virus infection suppression because they directly kill cells that have been infected with the virus [44]. CTLs emit various damaging compounds (perforins, granzyme, and FasL), and when the T cell receptor located in CD8^+^ T cells receives the specific message displayed by the MHC and peptide combination, it produces cytokines, such as interferon, TNF-α, and IL-2 [44]. As a consequence, the cell that enables virus multiplication is executed, thereby stopping virus propagation [45].

Viruses that cause long-term infections, such as HIV-1 and the Kaposi Sarcoma associated Herpes virus (KSHV), might elude immune surveillance by interfering with antigen presentation, which is necessary for immune activation, by decreasing the production of major histocompatibility complex I (MHC-I) molecules bound to the cellular membrane [46,47,48], as demonstrated in laboratory settings and living organisms [49]. Although these viruses have acquired the capacity to elude immune detection by interfering with antigen presentation, their underlying processes differ [49].

SARS-CoV-2’s open reading frame 8 (ORF8) produces a polypeptide that interacts with MHC-I molecules and causes their diminished expression [49]. Infected cells with the Wuhan strain and with the Delta variant have a reduced vulnerability to cytotoxic T lymphocyte destruction because MHC-I molecules are specifically selected for their degradation within lysosomes (autophagy) in cells that express the ORF8 protein. Another study found that the ORF6 protein from SARS-CoV-2 disrupts the stimulation of MHC-I genetic expression [50]. The virus suppresses the production of MHC type I in epithelial cells following virus infection, according to gene expression analyses in COVID-19 sufferers. Because MHC-I is essential for antiviral immunity, this mechanism has been identified as a potential instrument for virus immunological escape. These findings suggest that during SARS-CoV-2 infection, defective stimulation of MHC class I genetic expression in airways and epithelial cells from the intestines impairs cellular immunity mediated by CD8 T cells, increasing the likelihood of viral load intensification and extended disease [50].

The effect of Omicron genetic changes on the cellular immunological response produced by vaccination was recently investigated, both for CD4^+^ and CD8^+^ T cells by immune sequencing and epitope mapping [51]. It was observed that genetic changes in the Omicron variant were not selected to evade cellular immunity, and may not abrogate antigen presentation, as they only affect 21% of the class I-mediated cellular immunological response elicited by SARS-CoV-2 vaccines and 33% of the class II type. Because the genetic changes are clustered in the spike gene and are relatively scarce in the remaining genes, the impact of the Omicron variant is minimal (less than 5%) on the T cell responses directed to regions different from the spike caused by natural disease [51]. Results from this study suggested that the Omicron variant does not induce MHC-I destruction mediated by the ORF8 protein, and therefore it might not impair antigen presentation, as was the case for the Wuhan strain [49]. Even though current variants avoid the majority of the vaccine-induced neutralization, the currently offered SARS-CoV-2 vaccines have been shown to be effective. This could be because such vaccines elicit a stronger T cell reaction than predicted, which has helped vaccine recipients avoid severe (or even symptomatic) disease, supporting the hypothesis that the genetic changes in Omicron have only a small impact on the T cell immune response [51].

### 2.3. SARS-CoV-2-Driven Inhibition of Interferon Synthesis

One of the earliest mechanisms of protection against viral infection is the natural interferon (IFN) response [52,53]. Coronaviruses evade the host immune response by using different mechanisms, which include inhibition of IFN communication, antagonizing IFN synthesis, and boosting IFN tolerance [54,55,56]. Specifically, the SARS-CoV-2 infection produces an inadequate and postponed IFN-I response that has been considered the main contributor to disease severity [56,57,58,59,60,61,62,63,64].

In fact, it was discovered that several proteins of this pathogen can inhibit the IFN defensive response [65], resulting in altered IFN-I generation and impaired signal transduction [57,61,66,67,68]. The antiviral type I interferon response is repressed by the SARS-CoV-2 nucleocapsid protein by making direct contact with the mitochondrial antiviral-signaling protein (MAVS) [69,70,71]. Retinoic acid-inducible gene I (RIG-I-like receptors), which primarily identifies viral RNA, initiates signal transduction through MAVS, which serves as a central connector. Then, both post-transcriptional and post-translational processes regulate MAVS expression and activity, with ubiquitination and phosphorylation playing the most critical roles in modifying MAVS function [69,70,71]. A study found that RIG-I and MAVS interaction was blocked by the viral ORF9b protein from SARS-CoV-2, while the viral ORF7a protein disturbed the TBK1 protein, resulting in reduced IRF-3 phosphorylation, which is essential for type I IFN production [72].

Upon entering human lung cells, SARS-CoV-2 generated significantly more viruses but markedly lower IFN synthesis and inflammatory cytokines than SARS-CoV-1, explaining COVID-19’s asymptomatic transmission and delayed onset of sickness [73]. Furthermore, SARS-CoV-2 evades immunological recognition in alveolar macrophages by inhibiting endogenous IFN production, indicating that the immune system is not properly alerted during the early stages of the disease, allowing the pathogen to proliferate so that significant lung tissue destruction occurs [74]. The failure of alveolar macrophages to recognize the virus may be related to the disease’s reduced symptom phase, as the initial stage of COVID-19 disease is characterized by a scarcity of symptoms, permitting the development of viremia in infected persons, and potentially creating enormous difficulties with viral dissemination in the general public [74].

According to the previous studies, very low concentrations of IFN-β protein were reported in a considerable percentage of COVID-19 blood samples [75]. This validated previous investigation revealed lower amounts of IFN-α in hospitalized COVID-19 patients versus controls [63] and in severely ill COVID-19 individuals versus mild-moderate COVID-19 patients [62,64]. Research that assessed the existence of autoantibodies targeting type I IFNs in COVID-19 patients revealed the relevance of category I IFN-mediated immunity [60]. Remarkably, auto antibodies against IFN-α, IFN-ω, or both were found in 101 of 987 (10.2 percent) patients with mortal symptoms [60]. These auto antibodies, on the other hand, were discovered in only 4 of the 1227 (0.33 percent) people in the wider public, and in 0 of the 663 individuals without symptoms or moderate SARS-CoV-2 infection. These antibodies neutralized high amounts of category I IFNs, inhibiting SARS-CoV-2 infection in vitro [60]. Auto antibodies were detected in serum samples taken from certain patients prior to SARS-CoV-2 exposure. Individuals with such auto antibodies had very low concentrations of serum IFN-α throughout the acute phase of illness [60]. Importantly, neither the patients who had auto antibodies to category I Interferon nor alterations in genes implicated in the category I IFN cascade had previously suffered a dangerous virus disease prior to COVID-19. These findings show that category I IFNs could have a more essential contribution in defending the host from SARS-CoV-2 infection than against similar viral infections [60,76].

An insightful hypothesis has been offered on how postponed but enhanced type I IFN response leads to COVID-19’s grave progression by causing hyper-inflammation. Given the potential for IFN responses to worsen hyper-inflammation, utilizing category I or III IFNs as a therapy for COVID-19 patients should be carefully considered, particularly for individuals with advanced disease phases [77].

### 2.4. SARS-CoV-2 Induces Incomplete Mitophagy to Avoid Apoptosis of Some Infected Cells and to Increase Virus Replication

For viral survival and replication, virus-driven reconfiguration of cellular metabolism is a crucial tactic [78]. Many viruses, including SARS-CoV-2, have developed complex ways to manipulate numerous host functions for their benefit, such as modifying the cellular metabolism and altering the immune response [78,79,80,81,82]. Apoptosis is an essential element of cell reactivity to injury, which performs several important functions in homeostasis and development [83]. Following a viral infection, many cells may undergo programmed cell death, which might interfere with the creation and release of the offspring virus. Therefore, it was not unexpected to find that some viruses have evolved various strategies to prevent cell host apoptosis and increase their replication [83]. However, apoptosis induction has also been reported. Thus, both induction and inhibition of apoptosis can be activated by SARS-CoV-2 and might depend on the infected cells. For example, contemporary research found that a subset of infected T cells exhibited mitochondrial dysfunction and apoptosis, which could explain the leukopenia observed in severe COVID-19 cases [84]. On the contrary, a significant anti-apoptotic pathway controlled by SARS-CoV-2 is the nuclear factor kappa B (NF-κB) pathway. Apoptosis inhibitors are increased when NF-κB is induced. This approach is essential for viral infection, survival, and inflammation [85].

The mitochondria play several important roles in the cell’s life and death [86,87]. In response to viral infection, the function and structure of mitochondria change, impacting the efficacy of the immune response [88,89]. Pattern recognition receptors (PRRs), such as RIG1-like receptors (RLR), identify viruses as a key component of the antibody immunological response [70]. This identification activates interferon and pro-inflammatory cytokines with intrinsic antiviral function or specialized immune cells via signaling cascades with strong mitochondrial involvement [70].

RLRs also “trigger the mitochondrial antivirus signaling complex (MAVS), an external mitochondrial membrane protein group implicated in antivirus defense, by promoting the transcription of category 1 interferon” [90]. Viruses, as obligatory parasites, depend on cellular-molecular infrastructure and bioenergy production [91]. Viruses also regulate cellular functions and metabolic pathways in an attempt to utilize them for multiplication while simultaneously escaping the immunological response of the host cell. As a result of their importance in antiviral immune response, mitochondria are a prime target for viral regulation [92,93].

SARS-CoV-2 hijacks and manipulates mitochondrial function in infected cells to suppress antiviral response and host immunity [94,95]. According to certain in vitro findings [96], SARS-CoV-2 attaches to sections of the mitochondrion membrane TOM70 (mitochondrion import receptor subunit 70) protein and thus affects the cell’s class I interferon response. MAVS-regulated innate antiviral communication necessitates oxidative phosphorylation as well, as evidenced by cells with oxidative phosphorylation defects that generate fewer effective antiviral host defenses and induce interferon and cytokine release [97]. Another study discovered that the SARS-CoV-2 nucleocapsid protein blocks the antiviral class I interferon response by engaging in direct interaction with the MAVS protein [70].

Autophagy triggered during viral infections can either enhance or inhibit virus proliferation depending on the kind of virus and the related host cell [98,99]. Autophagy is inhibited by some viruses, including human cytomegalovirus, coxsackievirus B3, and herpes simplex virus type 1 [100,101,102]. Other viruses, such as the hepatitis B virus, the human immunodeficiency virus, the dengue virus, and the influenza virus, however, promote their replication and maturation by increasing autophagy [103,104,105,106]. The interplay of several coronaviruses (CoV) infections with autophagy has lately gained a lot of interest. Different investigations have raised two significant concerns: whether and how CoV infection affects autophagy, and whether or not autophagy is involved in CoV replication [99]. In this regard, research revealed that the SARS-CoV-2 membrane protein (M) triggered mitophagy, which decreased the class I IFN response, and that preventing mitophagy with particular inhibitors (3-MA and Wortmannin) suppressed viral replication [107]. Reduced oxidative phosphorylation is known to promote mitochondrial ROS (mtROS) generation in LPS-activated macrophages [108]. The SARS-CoV-2 ORF3a protein causes mitochondrial damage and mtROS liberation to enhance hypoxia-inducible factor 1 (HIF1-α) production, which enhances SARS-CoV-2 infection and cytokine release [109]. Furthermore, SARS-CoV-2-mediated activation of the HIF1-α pathway also induces a variety of downstream effects, including modification of the host cellular metabolism, inhibition of interferon synthesis, increased viral replication and inflammation [110,111,112,113,114,115].

However, if mitochondria are destroyed, the cell will not be able to perform other vital functions, such as the generation of energy through glycolysis or oxidative phosphorylation, so the term “mitophagy” is not correct. Therefore, the SARS-CoV-2 could impair only those selective mitochondrial pathways, which are related to the immune response, such as the inhibition of interferon synthesis. Recent work demonstrated that SARS-CoV-2 infection results in an incomplete autophagy response with an impaired capability to build double-membrane autophagosome vesicles, which are essential for successful SARS-CoV-2 replication [116]. Such incomplete autophagy was characterized by an increased autophagosome production, but blockage of autophagosome maturation to avoid lysosomal degradation [116].

In addition to the massive amounts of lipids, proteins, and RNA required to build hundreds of virions [117], it has been shown that oxygen utilization in latently infected cells is reduced, showing that oxidative phosphorylation is considerably decreased [118]. Glycolysis induction is essential for the latently infected cells to survive since blockage of this mechanism results in apoptosis. As a result, the switch to glycolysis produced by viruses may be important for virus survival, as well as nucleic acid duplication, virus production, and ejection [118]. A group of researchers discovered that an increased glucose concentration significantly promoted viral proliferation and inflammatory cytokine production, demonstrating the essential role of glycolysis in infected monocytes [119]. The SARS-CoV-2 replication requires glycolysis (the Warburg effect, a modified form of cellular metabolism commonly found in cancer cells, where instead of the oxidative phosphorylation, aerobic glycolysis originating from the upregulation of several major glycolytic enzymes is used to efficiently produce ATP), and SARS-CoV-2-induced mitochondrial ROS generation activates HIF1-α, which then potentiates the expression of glycolytic genes and IL-1b [119].

Furthermore, it was discovered that SARS-CoV-2-infected monocytes cause T cell malfunction and the death of epithelial cells in the lungs. These findings can explain why the uncontrolled serum glucose levels in diabetes patients are a determinant factor in complicated COVID-19 cases [119]. Interestingly, when only pyruvate is supplied to monocytes, SARS-CoV-2 replication is stopped, as this substrate supports the tricarboxylic acid (TCA) cycle and oxidative metabolism. Administrations of antioxidants, such as mitoquinol (MitoQ) or N-acetyl cysteine (NAC), were also shown to suppress virus multiplication in SARS-CoV-2-infected monocytes [119]. According to these studies, those antioxidants also blocked TNF-α, IL-6, INF-α, INF -β, and INF -λ, as well as HIF-α-target gene upregulation [119]. Inhibition of the HIF1-α pathway may be an effective strategy for the treatment of some patients who are suffering from acute respiratory distress. Interestingly, dexamethasone has been demonstrated to inhibit HIF1-α activity [120]. It is likely that, by blocking HIF1-α-induced glycolysis, dexamethasone inhibits SARS-CoV-2 replication.

### 2.5. Cell–Cell Infection and Immune Evasion through Tunneling Nanotubes

After newly generated viruses are discharged from the cell, the ACE2 attachment process is used to disseminate the infection, and emerging viruses merge their membrane with the cell membrane of the objective cell and then deliver their RNA inside the cytoplasm [121]. The new viral reproduction “sabotages” several intracellular vesicle machinery components. Often, the recently reproduced viruses are identified within bigger vacuolar ensembles and not as independent individual virus particles in the cytoplasm [122].

In this situation, antibody and cellular protection, in addition to a variety of other immune mechanisms, can disrupt the extracytoplasmic infection route. For instance, the immune system can degrade the virus and thus prevent the illness [123]. Several SARS-CoV-2 infection control techniques (such as immunization) depend on the host’s immunological defensive strategy against extra cytoplasmic virions [124]. Even when a considerable proportion of the population in numerous countries has been immunized, new symptomatic illnesses (breakthrough infections) in persons who were adequately inoculated have been documented, raising concerns regarding the creation of escape mutants [125]. Immunization reduces the seriousness of the illness, although SARS-CoV-2 can re-infect vaccinated individuals, resulting in dangerous sickness and even death [126]. Furthermore, the virus entrance pathways into permissive cells are unclear, and the way the pathogen spreads and multiplies in a subject with a powerful immunological response is still unknown [121].

The latest research has demonstrated that this virus produces infection to other cells using tunneling nanotubes (TNTs), which are nanometer- to micrometer-diameter cylindrical formations that connect adjacent or remote cells and promote cytoplasm communication between connected cells (Figure 2), enabling biomolecules to be shared and/or transported intra-cytoplasmically [121,127]. TNTs are defined as transitory formations that emerge and dissolve in a few minutes [128]. Viruses [129,130,131], prions [132] and other neurodegeneration-related prion-like proteins [133,134,135], such as tau [135,136], α-synuclein [137,138,139] and amyloid-β [140], as well as fungi [141] and mycoplasma [142] have all been found to help spread corresponding diseases using nanotubes.

In a related way to transport by the axons, TNT thickness increases when a large object is transported inside it. The cylindrical structure of some of these components has been successfully evidenced by scanning helium-ion microscopy (HeIM), demonstrating that TNTs can transport vesicles or viruses [121]. The role of TNT in a SARS-CoV-2 infection was explored in two recent papers [129,143]. TNT-mediated intracellular propagation has been suggested as a mechanism to hide viruses from T lymphocyte immune vigilance and antibody suppression in the extracellular fluid. TNTs can also reduce virus-cell connections, which might activate host defense mechanisms and antiviral responses. TNTs are used by several viruses, including influenza, HIV, and herpes simplex virus (HSV), to transfer their genetic material to new cells, permitting them to bypass host immune surveillance and pharmaceutical targeting [121].

SARS-CoV-2 may be added to the group of viruses that are able the spread and produce infection across host cells through TNTs, based on the findings from HeIM images [121]. As demonstrated by a recent study, SARS-CoV-2 infects non-permissive neurons by penetrating inside TNT (Figure 3) and traveling throughout this formation [127]. The entrance of SARS-CoV-2 to the infected cell is regulated mostly by the ACE2 receptor [38,144,145,146,147,148,149].

Despite being abundantly produced in the vascular endothelium of the alveoli [150], it was discovered at very low amounts in brain cells [151]. In human-derived neuronal cells (SH-SY5Y), other investigations did not detect the presence of the ACE2 receptor [127]. The primary objective of these researchers was to evaluate the characteristics and configuration of virions carried by TNTs, and also to study the processes that enable TNT-mediated virus transfer to non-receptive cells, for instance, as if contagious particles utilized TNTs as cellular connections to travel on the external surface of, or as conduits to be transported through the interior of the tube [127].

The virus is attached to the cellular membrane of TNTs generated among receptive cells, and vacuolar structures similar to viruses can be detected inside TNTs (Figure 3) produced among receptive cells and non-receptive cells, according to cryo-electron scanning [127,152]. These results hint at a possible novel strategy for SARS-CoV-2 propagation, one that can enable the virus to invade non-receptive cells while intensifying infection in receptive cells, allowing the virus to spread more widely throughout the host [121,127,152].

### 2.6. Cell–Cell Infection and Lymphocyte Cell Death through Syncytia Formation

Encapsulated viruses propagate in cellular cultures and tissues mainly through two mechanisms: cell-free particles and cell–cell close interaction [153,154,155,156,157]. Tight cell–cell interactions allow successful virus propagation to neighbor cells [156] and can establish synaptic connections, where there is a high viral load [154]. Surprisingly, cell–cell propagation exceeds the efficiency of cell-free propagation [158], and HIV and other viruses have been proven to utilize it in vivo [154,155,159]. For example, while the SARS-CoV-2 spike is better than the SARS-CoV-1 spike in inducing cell-to-cell propagation, the SARS-CoV-1 spike is superior at regulating cell-free propagation. Notably, this research also revealed that cell-to-cell transmission of SARS-CoV-2 can take place even when ACE2 is not present [152].

SARS-CoV-2 cell–cell propagation cannot be blocked by neutralizing antibodies in the serum from recovered COVID-19 victims, contrasting to cell-free infection [152,160]. The Beta variant of this virus is highly resilient to the vaccine serum blockade in cell-free propagation, but the Alpha version is more resilient to blockage by serum from vaccinated people in cell-to-cell propagation [152]. Cell–cell fusion can be caused by viral diseases, such as HIV, respiratory syncytial virus, and herpes simplex virus (HSV) [161]. SARS-CoV-2 infection is linked to syncytia production [162,163,164], which was found in histopathological lung tissues from people who died from COVID-19 [162]. Findings from a new study showed that SARS-CoV-2-infected Vero E6 cells can generate groups of many fused cells (syncytia) 24 h following infection. It is important to emphasize here that, in the uninfected cells, this type of fusion does not occur [121]. It has been proposed that virus-mediated cell fusion can increase viral genome transmission to nearby cells [163] via sharing cytoplasm [152].

SARS-CoV-2-induced syncytia generation has been linked to lymphopenia in the pulmonary tissue of individuals with critical COVID-19. Syncytia may select lymphocytes to be internalized by endocytosis and cause their death, resulting in lymphopenia [162,164]. Internalization, also known as entosis, is a non-apoptotic cell death process, in which swallowed cells are destroyed by lysosomal enzymes [165]. Lymphopenia is a distinguished signature of people with severe COVID-19, according to several reports, and it is linked to poor clinical outcomes. In corresponding cases, both CD4^+^ and CD8^+^ T cell numbers are decreased [166,167,168,169,170]. In very complicated COVID-19 cases, decreased B cell numbers are also reported [171]. Interestingly, there were fewer memory and regulatory T cells (Tregs) (CD25+) in the CD4^+^ T cell section, while the number of naïve T cells (CD45RA+) was elevated [172].

Further research demonstrated that people with COVID-19 had fewer airway Tregs than healthy controls, supporting the idea that Tregs depletion in the lungs directly relates to the severity of the condition. In complicated cases, the number of Tregs (which maintain immune equilibrium by inhibiting activation and inflammatory responses) was extremely low [173]. Tregs cells have previously been shown to reduce the cytokine storm caused by respiratory viruses, as well as virus-induced pneumonia and acute lung injury [174,175]. Because CD8 T cells help in the elimination of viruses from the lungs, they may potentially play an essential role in such pathologies [176]. Regulatory T cells can organize the initial enrollment of CD8 T cells towards the lungs and air passages, but they can similarly reduce their cellular response intensity and potential to release TNF-α, thereby reducing pathogenicity [177]. Likely, a decrease in Tregs numbers (owing to syncytia formation) weakens the immune system’s capacity to inhibit the excessive response, resulting in the cytokine storm [177].

As the cleavage of the S1/S2 is important for SARS-CoV-2-mediated syncytia generation and infection of lung cells [178], such characteristics can explain why Delta infection produces more syncytia, and its spike is more fusogenic due to effective cleavage of S1 and S2 [179,180]. Importantly, in contrast to the Delta variant, Omicron does not promote syncytia formation, according to recent studies [181,182,183,184,185]. Other contemporary research has demonstrated that Omicron replication is significantly reduced in human lung cancer cell lines (Calu3) and colorectal adenocarcinoma cells (Caco2). When compared to the wild-type strain and the Delta (B.1.617.2) variant, Omicron proliferation is significantly reduced in both the upper and lower respiratory system of infected transgenic mice expressing human ACE2 receptor (K18-hACE2), culminating in markedly reduced lung damage. In comparison to wild-type SARS-CoV-2 and the Alpha (B.1.1.7), Beta (1.351), and Delta variants, Omicron infection results in the smallest weight loss and the lowest fatality percentage [186].

These pieces of evidence, together with epidemiological data, show that infection with the Omicron variant causes less severe disease in humans [187]. This is also in line with the clinical course being less grave, as a decreased replication rate of Omicron was also described in lung epithelial cells as recently reported [188,189]. In vitro studies suggested that the Omicron variant cannot spread easily by cell fusion when compared to other variants, and adds to the evidence that the virus itself may produce a milder disease in animal models [189]. This is relevant because it restricts the virus replication to the higher breathing tissues, where cell-free propagation (that is independent of transmembrane serine protease 2 (TMPRESS2) supports infection and makes the virus migration to the lungs less probable, where cell merging is more important for viral propagation, and disease impact is more severe [181,186]. According to recent studies, Omicron infection causes reduced pulmonary illness in hamsters and mice that express the human ACE2 receptor [181,184,188,189,190].

### 2.7. Immune Evasion through Exosome Release

Exosomes are extracellular nanovesicles that form from intraluminal vesicles during endocytosis and are released from the cell surface. They are selectively equipped with a range of biological components, such as RNAs, proteins, and lipids. Originally, they were considered a way for the cell to dispose of unnecessary materials [191]. However, exosomes have been discovered to serve as carriers of biologically active molecules and perform several essential functions in a range of biological processes, including immune control and cellular communication [191]. In the case of disease, exosomes can contain viruses, virus proteins, and viral genetic material. Furthermore, exosome packaging has been shown to shield viruses from antibody inactivation and also permitted virus integration within the cells that do not contain viral receptors and would normally be immune to the pathogens [192]. For example, HIV-1 employs exosome surface properties to promote quick delivery of progeny virus and to hide the virus from immunological monitoring. Hence, HIV-1 can speed up the infection and spread by surrounding itself with exosomes [193].

Herpes simplex virus 1 [194], hepatitis E virus [195], hepatitis C virus [196], hepatitis B virus [197], and enterovirus 71 [198,199] all interact with exosomes and similarly evade antibody inactivation. Previous research has revealed that infected cells can discharge viral molecules that behave as fake targets for the immunological system. Hepatitis B virus (HBV) emits sub-viral particles, which are non-contagious elements containing the HBV virus merging glycoprotein, HBsAg, and exceed contagious virions by a factor of 2000:1 [200,201]. As a result, neutralizing antibodies (nAbs) attach to the more abundant sub-viral particles instead of attacking the infectious virus, allowing the virus to escape neutralization and infect cells [200].

By observing exceptionally slim segments of SARS-CoV-2 infected cells from the respiratory epithelium with electron microscopy, researchers discovered the existence of cytoplasmic membrane-bound vacuoles transporting particles similar to viruses [170]. Numerous vacuoles harboring many minute particles similar to viruses were identified in a cell from the glomerulus, and a nearby cell from the tubular epithelium [202]. Trojan vesicles have been proven to enter the extracellular fluid as a complete piece in high-resolution transmission electron micrographs exhibiting one or two viral particles within a very tiny vacuole of 200–250 nm in diameter [203]. Exosomes behaving as Trojan horses may explain the resurgence of virus RNA in individuals sick with COVID-19 [204]. Exosomes play a role in SARS-CoV-2 infection [205,206,207], and they could be used as indicators of illness seriousness [208]. RNA from this virus was found in the exosomal contents of the serum of COVID-19 patients but not in the serum of healthy participants, showing that this pathogen can spread through endocytosis [209].

Exosomes generated by virus-infected cells can negatively influence the immune response, and, according to most researchers, they affect it to benefit the increased viral proliferation [192]. Exosomes from hepatitis B virus-infected cells, for example, lowered the number of invading mononuclear cells in mice, implying that exosomes from infected cells can dampen the antibody immune response [210]. Viruses can also elude the immune system by altering the composition of exosomes. The HIV-1 protein Nef, for example, has been demonstrated to have low CD4 levels in exosomes produced by HIV-infected CD4^+^ T cells and to diminish the potential of the exosomes from CD4^+^ T cells to suppress HIV infection [211]. Exosome membranes also serve an important function in protecting viral cargo from destruction by host enzymes, as well as having other “smart” properties, such as high biocompatibility, capacity to surmount biological barriers, and low immunogenicity [212,213].

One of the viral defenses against antibodies could be the lowered efficacy of neutralizing antibodies (nAbs) during infection. Due to the relevance of neutralizing immunoglobulins for preventing illnesses, viruses developed highly complicated and clever strategies to nullify innate immunity [214]. Exosomes harboring the SARS-CoV-2 spike react with the secretory immunological system, impairing the efficacy of serum nAbs in inhibiting viral infection in serum from convalescent patients. These exosomes can be used as fake targets, extending the viral arsenal of exosome-based techniques for avoiding host immune response [214].

## 3. Conclusions

The fight between viruses and their hosts has an old history, and while the immune system has evolved to defend from these pathogens, viruses have acquired clever evasion mechanisms to avoid being detected and destroyed by the immune system [215]. SARS-CoV-2 has demonstrated a remarkable ability to evade antibody neutralization [34,216,217,218], jeopardizing vaccination efficacy [216,219,220]. Several evasion mechanisms are used by SARS-CoV-2 to avoid recognition by the immune system (Figure 4).

Sabotage of the mitochondria is one of the immune evasion mechanisms utilized by viruses where, by triggering the generation of the reactive oxygen species (ROS), mitochondrial physiology is impaired, and the interferon antiviral response is inhibited [109]. Seminal studies have revealed an intra-cytoplasmic route of infection in SARS-CoV-2, which takes place through the creation of tunneling nanotubes, hence increasing infection and bypassing extracellular recognition [121,127,152,160]. Syncytia-mediated lymphocyte death is another way for this virus to avoid immunological response. In addition to serving as a mechanism for viral infection, syncytia creation gives rise to lymphopenia in the lungs: SARS-CoV-2 infection causes the genesis of syncytia, thus possibly exacerbating the lymphopenia reported in severe cases of COVID-19 [162,164]. Disrupting antigen presentation is another technique to avoid immune surveillance. In this case, SARS-CoV-2 infection causes decreased production of MHC-I molecules: the virus proteins encoded by SARS-CoV-2’s open reading frames 6 and 8 directly interact with MHC-I molecules and cause their down-regulation [49,50]. SARS-CoV-2-infected cells are highly resilient to destruction by cytotoxic T lymphocytes due to the fact that MHC-I molecules are specifically targeted for destruction within lysosomes [49]. This immune evasion strategy causes an impaired antiviral T cell response, which may result in worsening symptoms and a longer recovery time [168]. Finally, encapsulating viruses inside exosomes has been found to protect them from antibody neutralization while also enabling them to integrate into cells that would otherwise be immune to them [192]. Exosomes harboring SARS-CoV-2 spike react with immunoglobulins and impair the efficacy of serum nAbs in suppressing viral infection in convalescent patient serum [214]. Another exosome-based technique in the virus armament for avoiding an immune attack is exosomes carrying viral envelope glycoproteins and serving as fake targets [214].

To summarize, the evidence suggests that SARS-CoV-2 employs the various immune evasion mechanisms described in this work, not only to avoid neutralizing antibodies and immune monitoring but also to impair interferon production and alter cellular immunity, which may explain why patients do not develop long-term protection against this elusive pathogen. Data analyzed in this work also shows that, compared to the Wuhan strain or the Delta variant, Omicron has developed mutations that have impaired its ability to generate syncytia, thus reducing its pathogenicity. Conversely, other mutations have allowed it to escape antibody neutralization and preventing cellular immune recognition, making it the most contagious and evasive variant to date.

## Figures and Tables

**Figure 1 biomedicines-10-01339-f001:**
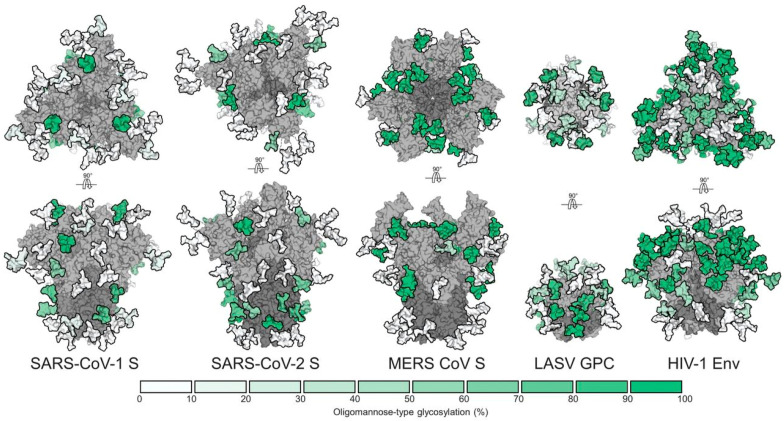
Glycan shields from different viruses. The number of glycans in SARS-CoV-2 is lower compared with the other viruses. This image is reproduced from an open-access article distributed under the terms of the Creative Commons CC BY license, which permits unrestricted use, distribution, and reproduction in any medium, provided the original work is properly cited. Source: [24].

**Figure 2 biomedicines-10-01339-f002:**
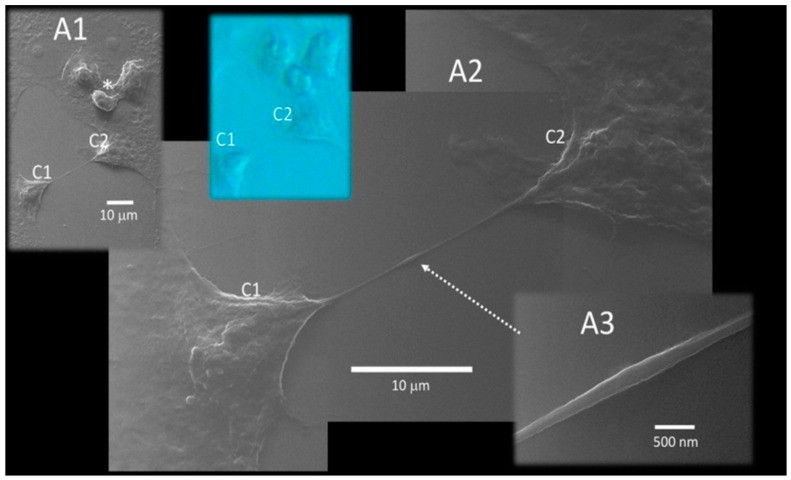
Tunneling nanotube (TNT), interconnecting two distant cells, C1 and C2. A trio of bulging and detaching infected cells (asterisk in A1) may be detected nearby, as well as healthy cells. Both cells are shaped with a stretched morphology by this extended TNT (A2). TNT stretched between two cells frequently bulges in the middle (dotted arrow; magnified in A3). This image is reproduced from an open-access article distributed under the terms of the Creative Commons CC BY license, which permits unrestricted use, distribution, and reproduction in any medium, provided the original work is properly cited. Source: [121].

**Figure 3 biomedicines-10-01339-f003:**
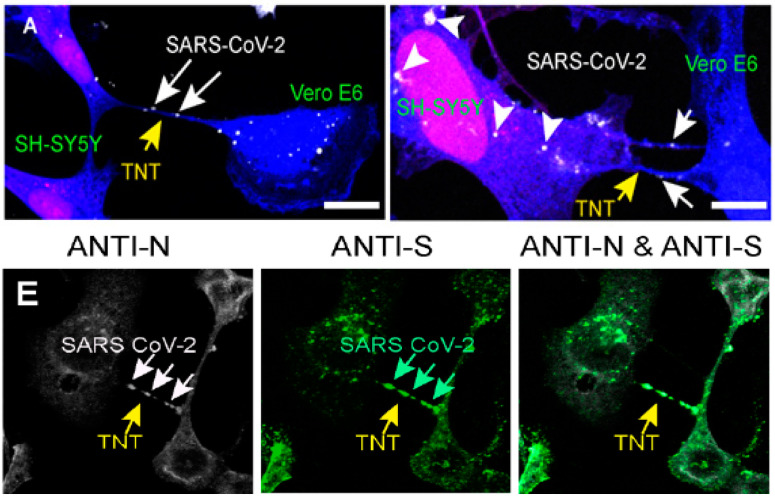
(A) SARS-CoV-2 can move from receptive infected Vero E6 cells to non-receptive SH-SY5Y mCherry cells thanks to TNTs. To detect the virus, co-cultures were fixed at 24 h (A left top) and 48 h (A right top) and stained with anti-N antibody. (E) SARS-CoV-2 was tagged using anti-nucleocapsid, anti-spike, and anti-N & anti-spike immunostaining. TNTs are indicated by yellow arrows among infected VeroE6 cells, whereas SARS-CoV-2 virions inside TNTs are indicated by white and green arrows. This image is reproduced from an open-access article distributed under the terms of the Creative Commons CC BY license, which permits unrestricted use, distribution, and reproduction in any medium, provided the original work is properly cited. Source: [127].

**Figure 4 biomedicines-10-01339-f004:**
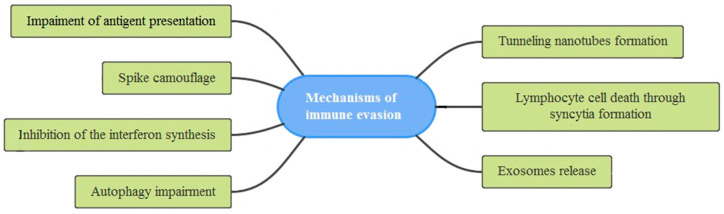
Schematic diagram showing the different evasion mechanisms developed by SARS-CoV-2 to avoid immune surveillance and attack by the immune system.

## Data Availability

Not applicable.

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
