# Peer review of "SARS-CoV-2: A Master of Immune Evasion"

_biomedicines, 2022, doi:10.3390/biomedicines10061339_

Round 1

Reviewer 1 Report

This is a comprehensive summary of the viral mechanisms (Covid-19) how it is capable bypassing the immune defense. of the host. The remark that using convalescent plasma, or specific antibodies might promote the escape of mutant viruses is correct, but the patients should be treated anyhow. 

Regarding the several abbreviations, I would like to suggest to add a glossary to the manuscript.

Author Response

This is a comprehensive summary of the viral mechanisms (Covid-19) how it is capable bypassing the immune defense of the host. The remark that using convalescent plasma, or specific antibodies might promote the escape of mutant viruses is correct, but the patients should be treated anyhow.

RESPONSE: Thank you for pointing this out. We rephrased this section of the manuscript to read: Recent investigations have demonstrated that COVID-19 patients treated with convalescent plasma showed significant improvement in clinical symptoms, a reduction in the level of viral antigens, and an increase in the blood oxygen saturation and lymphocyte ratio [9], although the use of convalescent plasma also promotes the production of antibody escape variants [11,12].

Regarding the several abbreviations, I would like to suggest to add a glossary to the manuscript.

RESPONSE: Thank you for pointing this out. The abbreviation list was added to the revised manuscript.

Reviewer 2 Report

Covid-19 pandemic has been persistent for more than two years around the world due to the unstopped appearance of viral mutants.  The Sars-CoV-2 mutant is now in its predominance to evade the world.  Comparing to Sars-Cov-1, this mutant has shown the increasing infectivity and decreasing fatality.  This manuscript summarizes the current knowledge on how Sars-CoV-2 could adopt seven strategies to escape our immune survey system.  In general, this review gave us a whole picture as wells as details of these seven strategies with extensive references cited.  In addition, the structure of this manuscript is clear and understandable which make it easy for readers to follow.

Minor: 

1) The abstract is not very informative.  It is better rephrase this part pointing out seven strategies which make Sars-CoV-2 more predominant.

2)  The authors should give us a diagram to summarize the strategies in the conclusion part. 

3) The writings should be improved

Author Response

Covid-19 pandemic has been persistent for more than two years around the world due to the unstopped appearance of viral mutants.  The Sars-CoV-2 mutant is now in its predominance to evade the world.  Comparing to Sars-Cov-1, this mutant has shown the increasing infectivity and decreasing fatality.  This manuscript summarizes the current knowledge on how Sars-CoV-2 could adopt seven strategies to escape our immune survey system.  In general, this review gave us a whole picture as wells as details of these seven strategies with extensive references cited.  In addition, the structure of this manuscript is clear and understandable which make it easy for readers to follow.

RESPONSE: We are thankful to this reviewer for appreciation of our work and glad that s(he) has found that our manuscript is easy for readers to follow.

Minor: 

1) The abstract is not very informative.  It is better rephrase this part pointing out seven strategies which make Sars-CoV-2 more predominant.

RESPONSE: Abstract includes description of the seven strategies used by SARS-CoV-2 to evade immune system.

2)  The authors should give us a diagram to summarize the strategies in the conclusion part. 

RESPONSE: Thank you for this great suggestion. We created such a picture and added it to the revised manuscript as new figure 4.

3) The writings should be improved

RESPONSE: We carefully checked and edited the manuscript.